# Retrospective Study of Clinicopathological Changes and Prediction Model for Canine Vascular Neoplasms

**DOI:** 10.3390/vetsci11050189

**Published:** 2024-04-26

**Authors:** Jidapa Suphonkhan, Chananchida Klaymongkol, Wijittra Khomsiri, Jedsada Wanprom, Saharuetai Jeamsripong, Narisara Chimnakboon, Anudep Rungsipipat, Araya Radtanakatikanon

**Affiliations:** 1Faculty of Veterinary Science, Chulalongkorn University, Bangkok 10330, Thailand; 2Research Unit in Microbial Food Safety and Antimicrobial Resistance, Department of Veterinary Public Health, Faculty of Veterinary Science, Chulalongkorn University, Bangkok 10330, Thailand; saharuetai.j@chula.ac.th; 3Department of Pathology, Faculty of Veterinary Science, Chulalongkorn University, Bangkok 10330, Thailand; narisara.chi@chula.ac.th (N.C.); anudep.r@chula.ac.th (A.R.); 4Center of Excellence for Companion Animal Cancer, Faculty of Veterinary Science, Chulalongkorn University, Bangkok 10330, Thailand

**Keywords:** dog, CBC, clinicopathology, diagnosis, hemangioma, hemangiosarcoma, hemostasis, neoplasm, prediction model, serum chemistry

## Abstract

**Simple Summary:**

Canine hemangiosarcoma and hemangioma are neoplasms of blood vessel-lining cells commonly found in dogs. Most tumors are often asymptomatic; however, ruptures of the affected organs, especially the spleen, frequently lead to hemorrhage or even hemorrhagic shock before diagnosis. This retrospective study used blood as a practical specimen and routine assessments, including hematology, serum biochemistry, and coagulation profiles, to develop predictive models for the early detection of canine vascular neoplasms. Analyses revealed associations between anemia and lymphopenia with hemangioma diagnosis, while anemia, lymphopenia, and hyperfibrinogenemia were associated with hemangiosarcoma diagnosis. These findings emphasize the importance of integrating comprehensive laboratory data with clinical information to facilitate early diagnosis and management of these critical conditions.

**Abstract:**

Vascular neoplasms, including hemangiosarcoma (HSA) and hemangioma (HMA), are more common in dogs than other domestic animal species; however, comprehensive laboratory screening tests for early diagnosis are currently limited. The aims of this study were to investigate general signalments, anatomic locations, and clinicopathological abnormalities of dogs diagnosed with vascular neoplasms and to determine the diagnostic significance of these abnormalities. Retrospective data of dogs with HMA, HSA, and healthy dogs were analyzed. Dogs with HMA and HSA were seniors, with mixed breeds being most affected. HMA affected predominantly non-visceral sites, while HSA was more common in visceral sites, particularly the spleen. In multivariate model analyses, the odds of HMA diagnosis were 5.5 times higher in anemic dogs and 33.0 times higher in lymphopenic dogs compared to dogs without the abnormalities. The odds of HSA diagnosis were 42.5 times higher in anemic dogs, 343 times higher in lymphopenic dogs and 92.7 times higher in dogs with hyperfibrinogenemia compared to dogs without the abnormalities. The study suggested that these identified abnormalities were nonspecific and commonly observed in various chronic diseases, and hence their combination with clinical information, such as diagnostic imaging and histopathology, is important to facilitate a more precise diagnosis of canine vascular neoplasms.

## 1. Introduction

Hemangiosarcoma (HSA) and hemangioma (HMA), neoplasms of vascular endothelial cells, are more common in dogs than other domestic animal species [1,2]. While HSA arises from bone marrow-derived endothelial cell precursors and expresses malignant behavior, HMA is considered to be a benign neoplasm or vascular malformation [3,4]. The spleen is the most frequent primary location for canine HSA, followed by the heart, typically found in the right atrium or auricle. Other common sites include the dermis and subcutis [2]. Clinical features of vascular tumors are variable and depend on the anatomic locations. Visceral HSA affecting the spleen and liver generally manifests nonspecific signs ranging from weight loss, abdominal enlargement, exercise intolerance, weakness, and pale mucous membrane to more serious conditions, such as collapse and shock due to internal tumor rupture. Cardiac tumor rupture may be followed by intrapericardial hemorrhage that leads to right-side heart failure or circulatory collapse due to cardiac tamponade. While canine visceral HSA is a severe and high-mortality-rate disease, HSA involving the skin or subcutis shows different clinical manifestations and is generally not seen at the emergency clinic. The tumors may range from small discrete blood blister-like lesions to much larger, deeply seated, painful, bruised or bleeding masses [5,6]. Determining whether HSA lesions in the skin originate as primary tumors, metastasize from internal tumors, or are connected to a multicentric disorder becomes challenging when they are widespread.

Although splenic HSA is a common malignant neoplasm in dogs, the list of differential diagnosis for splenic nodules is broad. Dogs with splenic HSA may show subtle signs of inflammation and anemia as consequences of tissue necrosis and chronic bleeding from the mass. Thrombocytopenia, hypofibrinogenemia and increased fibrin degradation products (FDPs) have been reported as important clinical findings in canine HSA [7,8]. The incidence of disseminated intravascular coagulation (DIC) is associated with hemostatic abnormalities causing high mortality rates in tumor-bearing dogs [9]. Clinicopathological features of canine HSA are varied and need to be promptly investigated. Definitive diagnosis of HSA requires histopathology, albeit tissue sampling techniques sometimes lead to uncontrolled bleeding due to high vascularization of the tumor and occult coagulopathy secondary to paraneoplastic syndrome. Fine-needle aspirate cytology of suspected lesions is challenging because the neoplastic cells are usually obscured by hemodilution. Visceral HSA is a life-threatening tumor associated with serious emergency complications. The tumors often rupture, leading to hemoabdomen and hemorrhagic shock even before the diagnosis has been made [10]. Local invasion as well as multiple organ metastasis occur early in the disease process due to direct hematogenous spreading of the malignant endothelial cells.

Early diagnosis to surgically remove the bleeding tumors is necessary to reduce the risk of tumor seedings in the abdomen and impede hematogenous metastasis to visceral organs. However, comprehensive laboratory screening tests or biomarkers specifically for diagnosing canine HSA are currently unavailable. Because of these tumors’ histological association with vasculatures, hematologic assessment could potentially detect the early onset of neoplasia. The aims of this retrospective study were to investigate general signalments, including age, breed and sex, anatomic locations and clinicopathological abnormalities, in dogs diagnosed with vascular tumors and to determine the diagnostic value of these variables. Changes in these variables could contribute to the development of a predictive model for the early detection of canine vascular neoplasia.

## 2. Materials and Methods

### 2.1. Tumor and Control Dogs

We conducted a retrospective study using the data from the Hospital Information System (HIS) electronic medical records of the Small Animal Hospital, Chulalongkorn University during 2017–2023. A filter search for HSA and HMA was applied to recruit tumor dogs. Inclusion criteria of targeted dogs were those with a definitive diagnosis of HSA or HMA at any affected location confirmed by histopathology, which is a gold standard for diagnosis of HSA and HMA. In cases where the diagnosis could not be conclusively determined based solely on routine histopathology, immunohistochemical staining for factor VIII-related antigen (von Willebrand factor; vWF) was performed to confirm endothelial differentiation [11,12]. Case signalment, imaging and clinicopathological data were assessed. Clinicopathological data including complete blood count, serum chemistry and hemostatic profile before treatment were retrieved from these cases. The healthy group comprised 100 blood donor dogs visiting the CU VET Blood Bank, Small Animal Hospital, Chulalongkorn University during 2021–2022. The inclusion criteria of the control dogs were clinically healthy dogs with body weight more than 15 kg and no history of illness, regardless of age, sex, or breed.

### 2.2. Clinicopathological Data

#### 2.2.1. Complete Blood Count (CBC)

CBCs were performed using a Mindray BC-5000 Vet automated hematology analyzer (Shenzhen Mindray Animal Medical Technology, Shenzhen, China) for red blood cell (RBC) count, hemoglobin concentration, hematocrit, mean corpuscular volume (MCV), mean corpuscular hemoglobin (MCH), mean cell hemoglobin concentration (MCHC), platelet count, total white blood cell count (WBC) and differential white blood cell count.

#### 2.2.2. Serum Chemistry

Blood chemistry was analyzed using the Mindray BS-800M automated analyzer (Shenzhen Mindray Bio-Medical Electronics, Shenzhen, China) for sodium, potassium, calcium, phosphorus, alkaline transaminase (ALT), alkaline phosphatase (ALP), cholesterol, triglyceride, blood urea nitrogen (BUN), creatinine, total protein and albumin.

#### 2.2.3. Hemostatic Profile

Citrated whole blood samples were centrifuged at 2500× *g* for 10 min at room temperature (25 °C) to collect platelet-poor plasma for the hemostatic test using a Sysmex CA-500 automated analyzer (Sysmex Corporation, Kobe, Japan). Variables included thromboplastin time (PT), activated partial thromboplastin time (APTT) and fibrinogen concentration. The reference ranges used to define hemostatic abnormalities on PT, APTT and fibrinogen concentration were established based on the results obtained from the healthy group (mean ± 2 standard deviations (SDs)).

### 2.3. Statistical Analysis

Descriptive analysis was used to demonstrate the characteristics of demographic distribution of dogs with HMA and HSA including age, breed and anatomic locations. Kruskal–Wallis test with post hoc Dunn’s test was conducted for age, weight, RBC count, HCT, hemoglobin, WBC count, differential WBC count, ALT, ALP, cholesterol, glucose, BUN, creatinine, total protein, albumin, PT, APTT, and fibrinogen concentration to evaluate the differences among healthy, HMA, and HSA dogs. The data were categorized based on the presence or absence of abnormalities, using their reference intervals to identify values outside the limits.

Univariate logistic regression analysis was used to identify abnormalities significantly predicting the occurrence of HMA or HSA. The odds ratio (OR) was used to compare the likelihood of the neoplasms occurring with each abnormality relative to the reference category (absence of that abnormality). Any variable showing significance on univariate testing (*p* < 0.02) was selected as a candidate for the multivariate analysis. Forward selection and backward elimination were employed for multivariate logistic regression analyses as model selection techniques to identify the final model (*p* < 0.05). All statistical analysis was based on two-sided hypothesis testing, with a significance level set at a *p*-value < 0.05. Statistical analyses were performed with Stata 18 (StataCorp, College Station, TX, USA).

## 3. Results

### 3.1. Patient Signalment

A total of 178 dogs with a final diagnosis of vascular neoplasms confirmed by histopathology were identified using the database on HIS search. One hundred and three (57.8%) dogs were diagnosed with HMA, and 75 (42.1%) dogs were diagnosed with HSA. The mean ± SD ages of dogs diagnosed with HMA and HSA were 11.4 ± 4.1 and 11.5 ± 4.3 years, respectively. For dogs diagnosed with HMA, 54.4% were male and 45.6% female. For dogs diagnosed with HSA, 53.3% were male and 46.7% female. The most common breed diagnosed with HMA was mixed (47.6%), followed by golden retriever (11.6%). The most common breed diagnosed with HSA dogs was mixed (40.0%), followed by golden retriever (12.0%). Non-visceral sites (93.2%), including cutaneous, subcutaneous and dermis, were the most common location of HMA, and the spleen (49.4%) was the most common site of HSA (Table 1).

### 3.2. Clinicopathological Abnormalities in Dogs with Vascular Neoplasms

Three dogs from each tumor group lacked pretreatment data and were excluded from clinicopathological study. Data availability varied in individual case, and the sample size used for each analysis is indicated.

For RBC mass, significant differences between the groups were detected in all variables, including RBC count, hemoglobin concentration, and hematocrit, with *p* < 0.001. The RBC mass variables in dogs with HSA were significantly lower than the other groups (Figure 1a–c). Among the RBC indices, only MCHC showed a significant difference (*p* < 0.0001), yet the lower and upper quartile values remained within the reference interval (Figure 1d). A significant difference was detected in platelet count across the groups (*p* = 0.002), and the HMA group had significantly higher platelet count than the other groups (Figure 1e). Significant differences between the groups were detected in total WBC count, neutrophil count, and lymphocyte count (*p* < 0.0001) (Figure 1f,g). Dogs with HMA had significantly lower total white blood cell count than the other groups. Lymphocyte count for the healthy dogs was significantly higher than tumor groups (Figure 1g).

For serum chemistry, significant differences between the groups were detected in BUN (*p* < 0.0001), creatinine (*p* < 0.0001), ALP (*p* < 0.0001) and albumin (*p* = 0.02). Dogs in the HMA group had higher BUN level compared to other groups (Figure 2a). Dogs in tumor groups had lower creatinine and higher ALP compared to healthy dogs (Figure 2b,c). For hemostatic profile, significant differences between groups were detected in all variables (*p* < 0.0001). Dogs in the HSA group had higher PT and APTT values than other groups (Figure 3a,b). Dogs in both tumor groups had higher fibrinogen concentration than healthy dogs (Figure 3c). Based on data from the healthy group (n = 100), reference intervals of hemostatic profiles were established as PT 5.2–7.1 s, APTT 8.4–16.2 s and fibrinogen concentration 102–229 mg%.

### 3.3. Logistic Regression Analysis of Identifiable Canine Vascular Neoplasms Using Clinicopathological Variables

The variables that were selected for initial model analysis were RBC count, hemoglobin concentration, hematocrit, MCHC, total WBC count, neutrophil count, lymphocyte count, platelet count, PT, APTT, fibrinogen concentration, BUN, creatinine, ALP and albumin concentration. All variables were treated as categorical variables; therefore, anemia (identified by RBC count, hemoglobin concentration, and hematocrit), hypochromia, leukocytosis, neutrophilia, lymphopenia, thrombocytopenia, prolonged PT, prolonged APTT, hyperfibrinogenemia, increased BUN, increased creatinine, increased ALP and hypoalbuminemia were defined as values outside the reference limits. The absence of each abnormality was used as a reference for logistic regression analysis.

Based on univariate logistic regression, the presence of anemia (defined by RBC count, hemoglobin concentration and hematocrit), lymphopenia, hyperfibrinogenemia, and increased BUN level were positively associated with the occurrence of HMA. The presence of leukocytosis was negatively associated with the occurrence of HMA (Table 2). The presence of anemia, lymphopenia, prolonged PT, prolonged APTT, hyperfibrinogenemia, and increased BUN and creatinine levels were positively associated with the occurrence of HSA (Table 3). Based on multivariate analyses, the odds of HMA diagnosis were 5.5 times higher (OR = 5.5, *p* = 0.012) in the presence of anemia defined by RBC count and 33.0 times higher (OR = 33.0, *p* = 0.001) in the presence lymphopenia compared to those without the abnormalities (Table 2). The odds of HSA diagnosis were 42.5 times higher (OR = 42.5, *p* = 0.011) in the presence of anemia defined by hemoglobin concentration, 343 times higher (OR = 345.7, *p* = 0.001) in the presence lymphopenia and 92.7 times higher (OR = 92.7, *p* = 0.004) in the presence of hyperfibrinogenemia compared to those without the abnormalities (Table 2).

## 4. Discussion

This study aimed to investigate general signalments, anatomic locations and clinicopathological variables, including hematology, serum chemistry and hemostasis profile, of dogs diagnosed with vascular neoplasms. The study showed a prevalence of HMA and HSA among mixed-breed dogs, deviating from prior findings. A population-based study revealed that the highest estimated incidence rate of malignant vascular tumors of all types was found in the boxer [15]. Recently, the genotype group containing German shepherd dogs showed a notably higher rate of splenic HSA diagnoses, with a 75% positive predictive value for HSA diagnosis within this group [16]. Geographical differences could contribute to this variation in our study, possibly influenced by the rising population of mixed-breed dogs in the surveyed regions. Dogs affected by HMA and HSA showed no significant age differences and were predominantly observed among dogs older than 11 years. This aligns with previous studies indicating that the age group at higher risk of all types of vascular tumors was between 10 and 12 years old [15,17]. In our study, we did not observe a sex predisposition for the presence of HMA or HSA, consistent with previous reports [15,18]. However, in human studies, there was a suggestion of a potential predisposition of vascular tumors in females, indicating hormonal involvement. Additionally, vascular tumors and malformations were identified frequently in infancy [19,20].

While the specific cause of HMA remains unknown, a potential association between UV light exposure and the biological characteristics of glabrous skin, particularly at the lower abdomen of dogs with white or thin hair coats, in the development of the tumors has been suggested. Previous studies reported that HMA presented approximately 3.8% to 4.5% of all skin tumors, with a notably higher prevalence at 73% compared to cutaneous HSA, which represented 27% of cases [17]. Histopathologically, HMA exhibits diverse subtypes: cavernous, infiltrative, capillary, arteriovenous, angiokeratoma, granulation tissue type, spindle cell, and solar-induced, with the first two being the most prevalent [19,21]. Our observation (unofficial report and unpublished) revealed that HMA predominantly affected cutaneous and subcutaneous areas, with the cavernous subtype being predominant. However, the current literature lacks reports correlating histological subtypes with aggressive tumor behavior.

We found a high prevalence of visceral HSA, especially in the spleen. This aligns with other studies that recognize the spleen as a frequent primary site, strongly associated with hemoperitoneum [10,22]. Non-traumatic hemoabdomen aids in diagnosing splenic HSA following the “rule of two thirds,” indicating that about two thirds of ruptured canine splenic tumors are attributed to underlying HSA [23,24]. The signs of hemoabdomen warrant prompt attention and tumor assessment for diagnosis, possibly explaining the higher prevalence of diagnosed splenic HSA. It was interesting that cardiac HSA was not observed in this study, despite being commonly reported as a primary site for HSA in dogs [25,26]. Antemortem diagnosis of a cardiac tumor using cytology or biopsy is challenging, and necropsy typically serves as the final diagnostic tool in identifying the type of cardiac HSA. In this study, because the diagnosis is often presumptive, relying on clinical and diagnostic imaging results, along with the inclusion criteria of histopathological confirmation as HSA, cases of cardiac HSA may have been underestimated. Moreover, a retrospective study reported concurrent cardiac masses in 8.7% of splenic HSA cases and concurrent splenic HSA in 29% of cardiac HSA cases, with 42% showing evidence of metastasis elsewhere [27]. Our study potentially underestimated the presence of simultaneous splenic and cardiac HSA. Additionally, the splenic HSA, or even other visceral HSAs identified, might be indicative of metastatic disease originating from cardiac HSA.

In human medicine, low hemoglobin concentration is the primary indicator of anemia. Our study investigated the relationship between canine vascular neoplasms and anemia using multiple variables, and found an association defined by all RBC mass variables. Anemia has been reported as a common clinical finding in canine HSA, particularly in its visceral form. It is frequently accompanied by regenerative signs as a result of hemorrhage and microangiopathic hemolysis [2,28]. We did not observe a significant difference in total protein concentration between groups. Moreover, despite the HSA dogs having lower serum albumin than the HMA dogs, the presence of hypoalbuminemia was not associated with the occurrence of HSA. Concurrent hemorrhage might not be the primary cause of anemia in vascular neoplasia, and anemia in these cases could be influenced by various factors, including chronic inflammatory diseases [29]. We observed significantly higher neutrophil counts in HSA dogs and lower lymphocyte counts in both neoplastic groups. However, only lymphopenia is significantly associated with occurrences of vascular neoplasia. Neutrophilic leukocytosis is a common finding in visceral HSA and likely associated with a paraneoplastic syndrome, secondary to tumor necrosis or stress-induced excessive endogenous corticosteroid [2,30]. Therefore, this leukogram pattern is nonspecific and often seen in chronic diseases.

Thrombocytopenia has been reported in 30–60% of dogs with HSA and can be considered a potential prognostic parameter [8,10,31]. Splenic HSA associated with splenomegaly can lead to abnormal function of the spleen and splenic sequestration of up to 90% of the total platelet count, causing thrombocytopenia [28]. In this study, although the platelet count in dogs with HSA was significantly low, an association between thrombocytopenia with vascular neoplasia was not identified. This could be explained by the fact that the samples were combined non-splenic HSA and had minimal relevance to splenic sequestration or consumption secondary to hemorrhage. Hemostatic dysfunctions such as thrombosis, hypercoagulability and DIC are clinical complications commonly present in malignant vascular tumors. Defects in secondary hemostasis including prolongation of PT and APTT, increased FDPs, increased D-dimer and hypofibrinogenemia are present in approximately 50% of dogs with HSA [7,8,9]. This is likely a result of hemostatic consumption and indicative of overt DIC. In this study, PT and APTT in dogs with HSA were significantly increased compared to other groups and associated with HSA. However, it is important to note that the lack of data on FDPs and D-dimers poses a limitation, making it challenging to accurately assess coagulation status and identify the presence of DIC. Interestingly, dogs with vascular neoplasia had significantly higher fibrinogen concentration and the presence of hyperfibrinogenemia was predictive of the neoplasms. The presence of hyperfibrinogenemia indicates persistent inflammation induced by neoplasms involving proinflammatory gene expression and cytokine release within the tumor microenvironment. This process promotes angiogenesis and tumor dissemination, potentially leading to an increase in the rate of fibrinogen synthesis, exceeding its consumption [32,33].

Our study has some limitations in addition to those previously mentioned. A significant limitation was the lack of evaluation of concurrent underlying diseases in dogs with vascular neoplasms, which may impact CBC, serum chemistry and hemostasis profiles. Given that the majority of tumor-bearing dogs are of advanced age and often present with concurrent chronic illnesses, the observed clinicopathological abnormalities were likely influenced by this multifactorial context. The retrospective nature of the study resulted in missing clinical information, such as clinical presentation at diagnosis, tumor staging, and median survival time. This limits our ability to accurately evaluate the utility of laboratory abnormalities in predicting neoplasia diagnosis. Without data on disease stage or symptoms at diagnosis, particularly if most patients are at an advanced stage, laboratory findings cannot be comprehensively interpreted as predictive tools for early diagnosis. Another confounding factor is that nearly half of HSA originated from non-visceral sites, and analyzing all tumor locations collectively may obscure significant hematologic alterations. Moreover, in the case of HMA, which manifest as benign tumors, systemic manifestations may be subtle, with observed abnormalities potentially linked to underlying conditions that the animals likely had. To address these limitations, further studies should categorize neoplasms by specific anatomic locations or utilize specific markers to obtain clinically relevant insights into hematologic parameters associated with vascular neoplasms. In the logistic regression analysis, continuous data were transformed into categorical dichotomous variables (presence or absence of abnormalities). This transformation was performed to account for variations in hematologic analyzers and reference intervals used across laboratories, as well as to make data more practically applicable for clinicians. The unexpectedly high ORs observed in some variables, such as lymphopenia and hyperfibrinogenemia, were potentially due to complete separation of categorical data. This might be attributed to the highly effective and essential nature of the variables or the limited sample size in the analysis. The unusually large ORs might indicate that these predictors, whether used alone or in conjunction with other variables in a multivariate model, may not be very informative as it its performing exceptionally well. Therefore, the interpretation of the clinicopathological profile should be complemented by other clinical data, including imaging findings and histopathological examination, to ensure an accurate diagnosis of canine vascular neoplasms.

## 5. Conclusions

This retrospective study on canine vascular neoplasms utilized a noninvasive and affordable routine blood test to evaluate clinicopathological abnormalities associated with the diagnosis of canine HMA and HSA. We found that anemia defined by low RBC count and lymphopenia was associated with a diagnosis of HMA, whereas anemia defined by low hemoglobin concentration, lymphopenia and hyperfibrinogenemia was associated with a diagnosis of HSA. However, these abnormalities were nonspecific and are commonly found in various chronic diseases. Therefore, the laboratory data should be interpreted cautiously and supplemented by additional clinical information, such as diagnostic imaging and histopathology. Future research should focus on integrating other specific biomarkers and establishing cutoff values for these parameters to enhance the accuracy of vascular tumor diagnosis.

## Figures and Tables

**Figure 1 vetsci-11-00189-f001:**
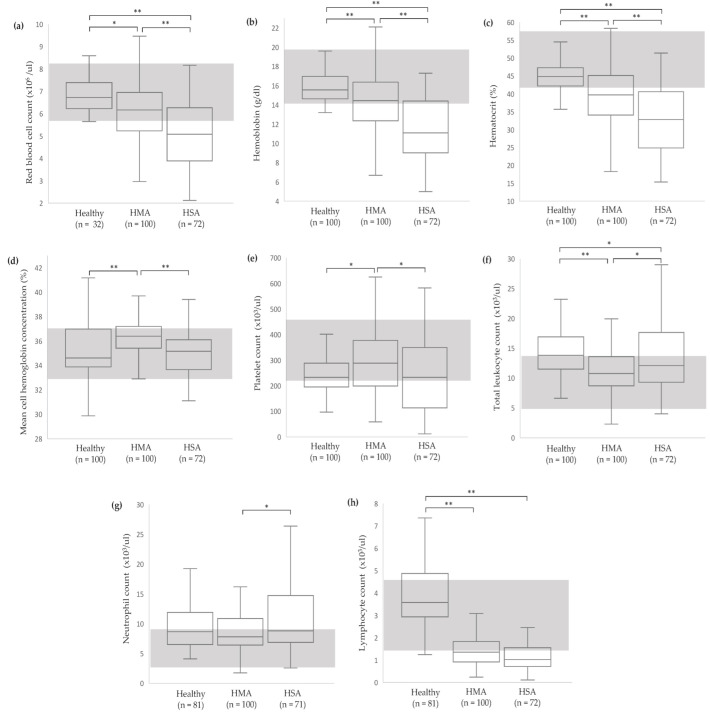
Box-and-whisker plots of CBC variables of healthy, HMA, and HSA dogs: RBC count (**a**), hemoglobin concentration (**b**), hematocrit (**c**), MCHC (**d**), platelet count (**e**), total WBC count (**f**), neutrophil count (**g**), and lymphocyte count (**h**). The central box represents the values from the lower to upper quartiles. The middle line represents the median. The vertical line extends from the minimum to the maximum value. The gray area indicates the reference interval [13]. * *p* < 0.05 and ** *p* < 0.0001 indicate significance. HMA, hemangioma; HSA, hemangiosarcoma.

**Figure 2 vetsci-11-00189-f002:**
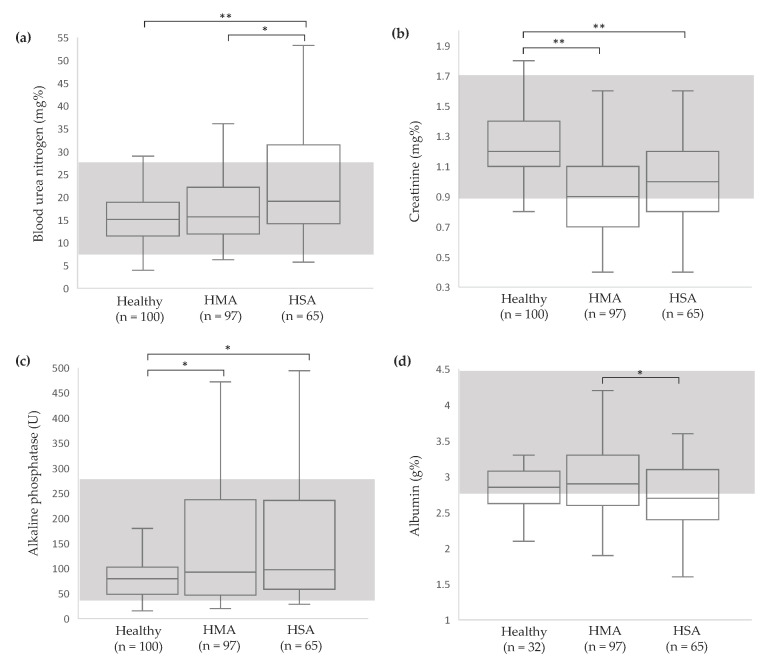
Box-and-whisker plots of serum chemistry variables of healthy, HMA, and HSA dogs: BUN (**a**), creatinine (**b**), ALP (**c**), albumin (**d**). The central box represents the values from the lower to upper quartile. The middle line represents the median. The vertical line extends from the minimum to the maximum value. The gray area indicates the reference interval [14]. * *p* < 0.05 and ** *p* < 0.0001 indicate significance. HMA, hemangioma; HSA, hemangiosarcoma.

**Figure 3 vetsci-11-00189-f003:**
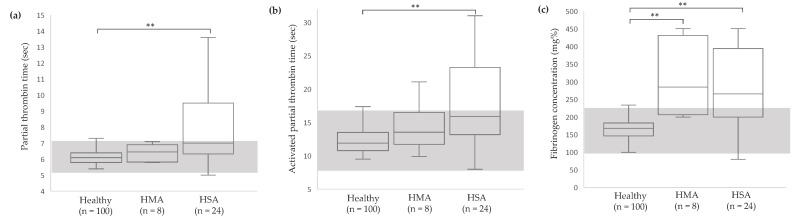
Box-and-whisker plots of hemostatic profiles of healthy, HMA, and HSA dogs: PT (**a**), APTT (**b**), fibrinogen concentration (**c**). The central box represents the values from the lower to upper quartile. The middle line represents the median. The vertical line extends from the minimum to the maximum value. The gray area indicates the reference interval obtained from the healthy group. ** *p* < 0.0001 indicate significance. HMA, hemangioma; HSA, hemangiosarcoma.

**Table 1 vetsci-11-00189-t001:** Distribution of sex, breeds, and tumor locations of dogs diagnosed with HMA (n = 103) and HSA (n = 75).

Signalments	Variables	No. (%) of Dogs
Sex	**HMA**	
	Male	56 (54.4)
	Female	47 (45.6)
	**HSA**	
	Male	40 (53.3)
	Female	35 (46.7)
Breeds	**HMA**	
	Mixed breed	49 (47.6)
	Golden retriever	12 (11.6)
	Poodle	10 (9.7)
	Labrador retriever	7 (6.8)
	Bangkaew	5 (4.8)
	Beagle	5 (4.8)
	Pomeranian	3 (2.9)
	Other ^1^	12 (11.6)
	**HSA**	
	Mixed breed	30 (40.0)
	Golden retriever	9 (12.0)
	Beagle	6 (8.0)
	Siberian husky	4 (5.3)
	Miniature schnauzer	4 (5.3)
	Poodle	4 (5.3)
	Pit bull	3 (4.0)
	Shih tzu	3 (4.0)
	Other ^2^	12 (16.0)
Anatomic locations	**HMA**	
	Non-visceral ^3^	96 (93.2)
	Spleen	7 (6.8)
	**HSA**	
	Spleen	37 (49.4)
	Non-visceral ^3^	33 (44.3)
	Omentum	3 (3.8)
	Liver	2 (2.5)

^1^ One each of shih tzu, Siberian husky, Chihuahua, pug, rottweiler, German shepherd, pit bull, cocker spaniel, Welsh corgi, French bulldog, mastiff and miniature schnauzer. ^2^ Two each of Bangkaew, Thai ridgeback, Jack Russell terrier, Labrador retriever and one each of Pomeranian, pug, Welsh corgi and dachshund. ^3^ Cutaneous and subcutaneous areas. HMA, hemagioma; HSA, hemagiosarcoma.

**Table 2 vetsci-11-00189-t002:** Association between the occurrence of HMA and significant predictors based on logistic regression analyses.

Variables	Initial Model ^1^		Multivariate Model ^2^	
	*p*-Value	OR ^3^ (95% CI)	*p*-Value	OR ^3^ (95% CI)
Anemia (by RBC)	**0.002**	7.00 (1.99–24.51)	**0.012**	5.49 (1.46–20.65)
Anemia (by HGB)	**<0.0001**	4.82 (2.42–9.61)	—	—
Anemia (by HCT)	**<0.0001**	4.12 (2.29–7.44)	—	—
Hypochromia	0.063	0.13 (0.01–1.11)	—	—
Leukocytosis	**<0.0001**	0.29 (0.16–0.56)	—	—
Neutrophilia	0.058	0.55 (0.30–1.02)	—	—
Lymphopenia	**<0.0001**	97.78 (13.09–730.57)	**0.001**	32.99 (4.28–254.40)
Thrombocytopenia	0.047	0.56 (0.31–0.99)	—	—
Prolonged PT	n/a	1	—	—
Prolonged APTT	0.072	5.26 (0.86–31.61)	—	—
Hyperfibrinogenemia	**<0.0001**	40.00 (6.98–229.30)	—	—
Increased BUN	**0.034**	4.76 (1.12–15.31)	—	—
Increased creatinine	0.980	1.02 (0.20–5.18)	—	—
Increased ALP	n/a	1	—	—
Hypoalbuminemia	0.811	1.12 (0.44–2.82)	—	—

^1^ Initial models were fitted for each variable. ^2^ Multivariate models were refitted using the significant variables identified in the initial models. ^3^ ORs indicate the change in odds when the variable is present. Variables significantly associated with the outcome (*p* < 0.02) are indicated in bold font. OR, odds ratio; CI, confidence interval; n/a, not applied; —, not assessed in the model. RBC, red blood cell count; HGB, hemoglobin concentration; HCT, hematocrit; PT, thromboplastin time; APTT, activated partial thromboplastin time; BUN, blood urea nitrogen; ALP, alkaline phosphatase. *p*-Value in bold font indicates statistical significance.

**Table 3 vetsci-11-00189-t003:** Association between the occurrence of HSA and significant predictors based on logistic regression analyses.

Variables	Initial Model ^1^		Multivariate Model ^2^	
	*p*-Value	OR ^3^ (95%CI)	*p*-Value	OR ^3^ (95%CI)
Anemia (by RBC)	**<0.0001**	18.17 (5.03–65.61)	—	—
Anemia (by HGB)	**<0.0001**	15.97 (7.43–34.31)	**0.011**	42.50 (2.39–754.14)
Anemia (by HCT)	**<0.0001**	15.5 (6.68–36.09)	—	—
Hypochromia	0.463	0.59 (0.15–2.39)	—	—
Leukocytosis	0.215	0.68 (0.36–1.26)	—	—
Neutrophilia	0.834	1.07 (0.65–2.04)	—	—
Lymphopenia	**<0.0001**	205.4 (26.74–1577.5)	**0.001**	345.66 (12.39–9653.14)
Thrombocytopenia	0.515	1.22 (0.67–2.24)	—	—
Prolonged PT	**<0.0001**	2.74 (6.76–111.23)	—	—
Prolonged APTT	**<0.0001**	9.43 (2.92–30.30)	—	—
Hyperfibrinogenemia	**<0.0001**	24.39 (6.67–86.43)	**0.004**	92.73 (4.41–1951.60)
Increased BUN	**<0.0001**	12.38 (3.47–44.13)	—	—
Increased creatinine	**0.009**	5.88 (1.55–22.27)	—	—
Increased ALP	n/a	1	—	—
Hypoalbuminemia	0.144	2.03 (0.79–5.23)	—	—

^1^ Initial models were fitted for each variable. ^2^ Multivariate models were refitted using the significant variables identified in the initial models. ^3^ ORs indicate the change in odds when the variable is present. Variables significantly associated with the outcome (*p* < 0.02) are indicated in bold font. OR, odds ratio; CI, confidence interval; n/a, not applied; —, not assessed in the model. RBC, red blood cell count; HGB, hemoglobin concentration; HCT, hematocrit; PT, thromboplastin time; APTT, activated partial thromboplastin time; BUN, blood urea nitrogen; ALP, alkaline phosphatase. *p*-Value in bold font indicates statistical significance.

## Data Availability

All data generated or analyzed during this study are included within the article.

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
