# Peer review of "Retrospective Study of Clinicopathological Changes and Prediction Model for Canine Vascular Neoplasms"

_vetsci, 2024, doi:10.3390/vetsci11050189_

Round 1

Reviewer 1 Report

Comments and Suggestions for Authors

The statistical analysis has been properly conducted. 

Authors emphasize the importance of an early diagnosis in HSA patients and propose some laboratory abnormalities to develop a predictive model for early detection of this still fatal disease: based on the data reported in the study, key informations, such as tumor stage and clinical compliant at diagnosis, are lacking, therefore limiting the possibility to establish the real usefulness of these laboratory abnormalities to predict the diagnosis of HSA. 

Information regarding stage of the disease and symptoms and diagnosis should be added: if most of the analyzed patients are at an advanced stage of disease, laboratory data such as anemia and DIC cannot be interpreted and used as predictive tools for an early diagnosis.

Authors should discuss another limit of the study, which is the lack of availability of coagulation factors nowadays commonly used to assess hypo or hypercoagulable states, including FDPs and d- Dimers.

The laboratory data were analyzed and categorized as dichotomous variables (presence or absence of abnormalities): authors should consider and discuss that, analyzing these data based not only on presence/absence but also on the severity of the abnormalities could change the significatively of the results, especially considering that, as stated by the authors, many abnormalities are not specific and common to several other diseases. 

LINE 15: “These neoplasms can be asymptomatic but deadly even in the early stages due to ruptured masses”. An early stage in visceral HSA is usually considered a stage I, which is typically a non- ruptured tumor, most of the cases diagnosed in the routine clinical practice and reported in the literature are usually stage II and III. Please rephrase the sentence (e.g. most of the tumors are diagnosed only after hemorrhage due to rupture of the organ).

LINE 43-44: this sentence sounds like a comparative one: if so, should be “more common than…”

LINE 50: substitute “affected” with “affecting”

LINE 50: change “range” with “ranging”

LINE 62: splenomegaly without focal lesions is unlikely in splenic HSA: substitute “splenomegaly” with “splenic nodules or masses” or at least add these features to the description.

Line 365: considering the inclusion criteria of an histological diagnosis, cases of cardiac HSA might have been underestimated, due to the fact that in most of the cases it is a presumptive diagnosis based on clinical and diagnostic imaging results.

LINE 437 : I would suggest to delete or rephrase  this sentence: “This retrospective study of canine vascular neoplasms provides valuable insights into clinicopathological abnormalities that potentially predict the diagnosis of canine HMA and HSA using a non-invasive, routine and affordable procedure”. Results of this study are inadequate to conclude that the described abnormalities can potentially predict the diagnosis of HSA and HMA and should interpreted cautiously.

Comments on the Quality of English Language

The article is generally well written, although needs English language editing

Author Response

Dear reviewer 1

We would like to extend our sincere gratitude for your valuable comments and kind suggestions on the manuscript. Your insights have greatly contributed to the improvement of this work, and we appreciate the time and effort you have dedicated to reviewing it. The following are point-by-point responses to the comments. Please refer to the attachment with track changes for details.

COMMENT 1: based on the data reported in the study, key informations, such as tumor stage and clinical compliant at diagnosis, are lacking, therefore limiting the possibility to establish the real usefulness of these laboratory abnormalities to predict the diagnosis of HSA. Information regarding stage of the disease and symptoms and diagnosis should be added

RESPONSE 1: We absolutely agree that this is probably the major limitation of this study. We stated it in DISCUSSION [Line 440-445]

COMMENT 2: Authors should discuss another limit of the study, which is the lack of availability of coagulation factors nowadays commonly used to assess hypo or hypercoagulable states, including FDPs and d- Dimers.

RESPONSE 2: We added this limitation in DISCUSSION [Line 413-415 ]

COMMENT 3: The laboratory data were analyzed and categorized as dichotomous variables (presence or absence of abnormalities): authors should consider and discuss that, analyzing these data based not only on presence/absence but also on the severity of the abnormalities could change the significatively of the results, especially considering that, as stated by the authors, many abnormalities are not specific and common to several other diseases. 

RESPONSE 3: This is a great suggestion. Initially, we tried this approach and found statistical limitations due to the nature of the data. Firstly, severity scales were not established for every variable, and we lacked sufficient data to do so. Additionally, the sample size in each group was too small to achieve significant statistical power. We also prior analyzed the data as continuous variables, which revealed almost similar result trend as dichotomous categorical variables. We then decided to transform the data to the categorical variable. This was done to accommodate the variation in hematologic analyzers used in each labs and the reference intervals used in individual clinics, making it more practical for clinicians to apply these values in their practice. We elaborated this point in DISCUSSION [Line 452-456]

COMMENT 4: LINE 15: “These neoplasms can be asymptomatic but deadly even in the early stages due to ruptured masses”. An early stage in visceral HSA is usually considered a stage I, which is typically a non- ruptured tumor, most of the cases diagnosed in the routine clinical practice and reported in the literature are usually stage II and III. Please rephrase the sentence (e.g. most of the tumors are diagnosed only after hemorrhage due to rupture of the organ).

RESPONSE 4: Revised. SIMPLE SUMMARY [Line 15-17]

COMMENT 5: LINE 43-44: this sentence sounds like a comparative one: if so, should be “more common than…”

RESPONSE 5: Revised. INTRODUCTION [Line 44] 

COMMENT 6: LINE 50: substitute “affected” with “affecting”

RESPONSE 6: Revised. INTRODUCTION [Line 54] 

COMMENT 7: LINE 50: change “range” with “ranging”

RESPONSE 7: Revised. INTRODUCTION [Line 54] 

COMMENT 8: LINE 62: splenomegaly without focal lesions is unlikely in splenic HSA: substitute “splenomegaly” with “splenic nodules or masses” or at least add these features to the description.

RESPONSE 8: Revised. INTRODUCTION [Line 67] 

COMMENT9: Line 365: Considering the inclusion criteria of an histological diagnosis, cases of cardiac HSA might have been underestimated, due to the fact that in most of the cases it is a presumptive diagnosis based on clinical and diagnostic imaging results.

RESPONSE 9: Revised. DISCUSSION [Line 371-377]

COMMENT 10: I would suggest to delete or rephrase  this sentence: “This retrospective study of canine vascular neoplasms provides valuable insights into clinicopathological abnormalities that potentially predict the diagnosis of canine HMA and HSA using a non-invasive, routine and affordable procedure”. Results of this study are inadequate to conclude that the described abnormalities can potentially predict the diagnosis of HSA and HMA and should interpreted cautiously.

RESPONSE 10: We truly agreed and emphasized this in CONCLUSION [Line 478-470, 474]

Thank you again for your invaluable contribution. We are looking forward to hearing from you and any feedbacks you may still have.

Sincerely, 

Authors

Reviewer 2 Report

Comments and Suggestions for Authors

Author Response

Dear reviewer 

We would like to extend our sincere gratitude for your valuable comments and kind suggestions on the manuscript. Your insights have greatly contributed to the improvement of this work, and we appreciate the time and effort you have dedicated to reviewing it. The following are point-by-point responses to the comments. Please refer to the attachment with track changes for details.

COMMENT 1: 14-23: The simple summary talks about canine hemangiosarcoma and hemangioma, but it needs to specify the location of the neoplasms, it is not recommendable to generalize that all the hemangiosarcoma and hemangiomas might lead to hemorrhagic shock since the cutaneous ones rarely cause hypovolemia.  

RESPONSE 1: We absolutely agreed that it was over generalized. Revised. SIMPLE SUMMARY [Line 15-17]

COMMENT 2: 24-38: why did the authors include all the visceral and non-visceral HAS and HMA in these hematologic results? Wouldn’t it be better to only focus on one location site (e. g. spleen)? Different location sites will cause different blood abnormalities since will compromise the function of specific organs, even when you simply classify visceral vs non-visceral sites, it is too broad to obtain reliable bloodwork results. 

RESPONSE 2: This is a great advise and we totally agree with it. We attempted to include as many cases as possible from all locations and analyzed them collectively to achieve significant statistic power. However, this approach resulted in unspecific findings. We have noted this limitation in DISCUSSION [Line 503-510].

COMMENT 3: 92-94: More details about the parameters to diagnose HSA and HMA should be included. This part is crucial and deserves reference(s) used to classify these vascular tumors. Do any immunohistochemical markers were used to distinguish HSA and HMA from other vascular tumors? 

RESPONSE 3We absolutely agree. The gold standard of diagnosis of HMA and HSA remains histopathology, but histomorphological overlapping with other sarcomas is definitely challenging. Based on pathology reports, in cases of ambiguity, IHC staining for von Willebrand factor (vWF) was performed to differentiate HSA from other sarcomas. For lymphangiosarcoma, which is also vWF positive, histomorphological features should be distinctive from HMA and HSA. However, this tumor is considered very rare. Texts were added MATERIALS AND METHODS [Line 99-108] and references [11,12] were included.

Comment 4: 153 and Table 1. The non-visceral sites include cutaneous, subcutaneous, and dermal areas. Based on Table 1, anatomic location section, it seems that the non-visceral HMA and HSA are only the cutaneous, subcutaneous, and dermal areas, but “dermal” means “skin”, so I do not know why the authors separate “cutaneous” from “dermal” 

RESPONSE 4: Thank you for pointing this out. These locations were initially identified based on clinical reports hence the words varied per clinician. Regarding anatomy of the skin, proper terms should be cutaneous and subcutaneous. Revised [Line 213]

Comment 5: 352. In the discussion, the cavernous is the predominant subtype of HMA, but it is not displayed in the results. 

RESPONSE 5: Due to the nature of retrospective data, we lacked complete pathology reports, including histologic subtypes of tumors in some cases. While it appeared that cavernous subtypes were more prevalent compared to others, we were unable to conduct an official evaluation using statistical analysis. Texts were added to elaborate this point. DISCUSSION [Line 384]. 

Comment 6: 370-372: This is not understood, what do you mean by “underestimated”? Does it sound like the authors place a possibility that the splenic or visceral HSAs might have originated from the heart (metastasis)? However, no cardiac HSA was seen in your data. 

RESPONSE 6: We did not observe a cardiac HSA in this study. We suspected that this was likely due to the inclusion criteria that only recruited cases with histologically confirmed HSA which is not typically feasible for cardiac tumors. Also, there could be cases with HSA in multiple locations but without lights from necropsy, it is difficult to accurately confirm if the splenic or visceral HSAs observed in this study metastasized from/to the heart. We clarified this in DISCUSSION [Line 396-405] 

Thank you again for your invaluable contribution. We are looking forward to hearing from you and any feedbacks you may still have.

Sincerely, 

Authors

Round 2

Reviewer 1 Report

Comments and Suggestions for Authors

I appreciate that the authors improved the manuscript based on reviewers' suggestions. 

Comments on the Quality of English Language

The paper can be accepted in this form, after minor editing of english language.